Early survey with bibliometric analysis on machine learning approaches in controlling COVID-19 outbreaks

Chiroma Haruna 1
Ezugwu Absalom E. 2 ezugwua@ukzn.ac.za
Jauro Fatsuma 3
Al-Garadi Mohammed A. 4
http://orcid.org/0000-0002-5511-1272 Abdullahi Idris N. 5
Shuib Liyana 6
1 Future Technology Research Center, National Yunlin University of Science and Technology , Yulin , Taiwan
2 School of Mathematics, Statistics, and Computer Science, University of KwaZulu-Natal , KwaZulu-Natal , South Africa
3 Department of Computer Science, Faculty of Science, Ahmadu Bello University , Zaria , Nigeria
4 Department of Biomedical Informatics, Emory University , Atlanta, GA , USA
5 Department of Medical Laboratory Science, College of Medical Sciences, Ahmadu Bello University , Zaria , Nigeria
6 Department of Information System, Universiti Malaya , Kuala Lumpur , Malaysia
Elazab Ahmed
Electronic publication date: 2020 Nov 23
Publication date: 2020
Volume: 6
Electronic Location ID: e313
Received 2020 Jul 26; Accepted 2020 Oct 15
Copyright: © 2020 Chiroma et al.
Copyright year: 2020
Copyright holder: Chiroma et al.
License: This is an open access article distributed under the terms of the Creative Commons Attribution License, which permits unrestricted use, distribution, reproduction and adaptation in any medium and for any purpose provided that it is properly attributed. For attribution, the original author(s), title, publication source (PeerJ Computer Science) and either DOI or URL of the article must be cited.
License URL: https://creativecommons.org/licenses/by/4.0/

Keywords: Bibliometric analysis, Convolutional neural network, COVID-19 pandemic, COVID-19 diagnosis tool, Machine learning

Funding: The authors received no funding for this work.

==============================
Background and Objective

The COVID-19 pandemic has caused severe mortality across the globe, with the USA as the current epicenter of the COVID-19 epidemic even though the initial outbreak was in Wuhan, China. Many studies successfully applied machine learning to fight COVID-19 pandemic from a different perspective. To the best of the authors’ knowledge, no comprehensive survey with bibliometric analysis has been conducted yet on the adoption of machine learning to fight COVID-19. Therefore, the main goal of this study is to bridge this gap by carrying out an in-depth survey with bibliometric analysis on the adoption of machine learning-based technologies to fight COVID-19 pandemic from a different perspective, including an extensive systematic literature review and bibliometric analysis.

Methods

We applied a literature survey methodology to retrieved data from academic databases and subsequently employed a bibliometric technique to analyze the accessed records. Besides, the concise summary, sources of COVID-19 datasets, taxonomy, synthesis and analysis are presented in this study. It was found that the Convolutional Neural Network (CNN) is mainly utilized in developing COVID-19 diagnosis and prognosis tools, mostly from chest X-ray and chest CT scan images. Similarly, in this study, we performed a bibliometric analysis of machine learning-based COVID-19 related publications in the Scopus and Web of Science citation indexes. Finally, we propose a new perspective for solving the challenges identified as direction for future research. We believe the survey with bibliometric analysis can help researchers easily detect areas that require further development and identify potential collaborators.

Results

The findings of the analysis presented in this article reveal that machine learning-based COVID-19 diagnose tools received the most considerable attention from researchers. Specifically, the analyses of results show that energy and resources are more dispenses towards COVID-19 automated diagnose tools while COVID-19 drugs and vaccine development remains grossly underexploited. Besides, the machine learning-based algorithm that is predominantly utilized by researchers in developing the diagnostic tool is CNN mainly from X-rays and CT scan images.

Conclusions

The challenges hindering practical work on the application of machine learning-based technologies to fight COVID-19 and new perspective to solve the identified problems are presented in this article. Furthermore, we believed that the presented survey with bibliometric analysis could make it easier for researchers to identify areas that need further development and possibly identify potential collaborators at author, country and institutional level, with the overall aim of furthering research in the focused area of machine learning application to disease control.

Introduction

The novel 2019 coronavirus disease pandemic emerged on December 12, 2019, and the Chinese government announced the isolation of new types of coronavirus on January 7, 2020 (Imai et al., 2020). This virus was code-named as COVID-19 by the World Health Organization (WHO) on January 12, 2020. However, it was renamed to Severe Acute Respiratory Syndrome Coronavirus-2 (SARS-CoV-2) by the international committee on the taxonomy of the virus (Gorbalenya, 2020). Since the pandemic, the number of confirmed cases as of October 10, 2020 reached 36,616,555 with 1,063,429 recorded fatalities. According to the origin of the first confirmed case report, the infection’s transmission was from animal to human, that is, zoonotic agent (Read et al., 2020). The virus has extended beyond China to other continents of the world (Wang et al., 2020a). The rise in the number of cases in Wuhan and other countries after the evacuation of the cases and closure of the marketplace in Wuhan shows a secondary transmission between humans. Related to the severe acute respiratory syndrome (SARS), the COVID-19 pandemic occurred during the China spring festival, the most famous celebration in China that attracted about 3 million Chinese people who traveled throughout the country. This festival period might have created an avenue for the transmission of this contiguous disease, making its prevention and control very difficult. The report on the secondary cases was made approximately 10 days after the first outbreak of the virus. These new patients were infected through human-to-human transmission because they never had any contact with the Wuhan marketplace, the origin of the first case of the pandemic (Read et al., 2020). The first non-Chinese infected case that spread over the provinces of China and then to the continents of Asia was reported on January 13, 2020, without any connection to the epidemiology at the market place in Wuhan (Hui et al., 2020). Subsequently, the spread continued, and various cases from countries abroad, such as the USA, Italy, and France began to be reported (Holshue et al., 2020).

Human-to-human transmission usually occurs in the case of close contact between humans. Restricting human-to-human spread is crucial for decreasing secondary infections among healthcare workers and close contact. The WHO recommended infection control interventions to reduce the risk of the general spread of acute respiratory diseases such as avoiding close contact with infected patients, frequent washing of hands in case of contact with ill people, and avoiding unprotected contacts with wild or farm animals. People suffering from the acute respiratory infection should always observe cough etiquette by maintaining distancing, covering their cough, sneezing with disposable tissues, and washing their hands frequently. Within the healthcare system, equipment improved standard infection control and prevention practices are recommended, especially in the hospital emergency unit (World Health Organization, 2020). Progressively, an interim clinical guideline has been established by the US Center for Disease Control and Prevention for the COVID-19 pandemic to place control measures and reduce the spread of SARS-CoV-2 in the USA (Patel & Jernigan, 2020).

In minimizing the devastating effect of the COVID-19 pandemic and fighting the virus, the timely processing of COVID-19 data analytics is highly critical. The data can come from different aspects such as the patients, fractionated into molecular, society, and population, which can help in the prevention and treatment of COVID-19 (Park et al., 2020). Machine learning is one of the major technologies in combatting COVID-19 (Elavarasan & Pugazhendhi, 2020; Kumar, Gupta & Srivastava, 2020). Much effort based on synergy from the machine learning and biomedical research community is ongoing to develop different approaches to combat the disease. Ongoing efforts include the screening of patients, detection and monitoring of patients, development of drugs, repurposing of drugs to treat patients with COVID-19, predicting the protein structure related to COVID-19, COVID-19 diagnostic systems, COVID-19 vaccine development, and analyzing personalized therapeutic effects for the evaluation of new patients (Alimadadi et al., 2020). The overall goal of these disease control techniques is to prevent the spread; track confirmed cases, recoveries, and mortality; and predict future pandemics (Vaishya et al., 2020).

Many studies adopted machine learning to fight the COVID-19 pandemic from a different perspective. A systematic literature review of published and preprinted reports on prediction models for diagnosing COVID-19 was reported. Prediction models were found floating the academic literature very rapidly to help in diagnosing COVID-19 and making critical medical decisions (Wynants et al., 2020). However, the major issue with the work is that it uses preprint publication that has not been validated by peer review. The review is not specific to machine learning for COVID-19. Sources of COVID-19 datasets that can guide researchers access different data for studies on COVID-19 are absent. The study is limited to the diagnosis and prognosis of COVID-19. A comprehensive taxonomy on the prediction tools to show the area that needs improvement is missing in the survey. Bibliometric analysis is not integrated into the study. Similar bibliometric analyses have been reported in the literature as presented by Chahrour et al. (2020), Hossain (2020), and Lou et al. (2020). However, these existing analyses differ from the current bibliometric analysis in this study because the current analysis result focuses on the application of machine learning techniques to combat COVID-19 pandemic as opposed to various literature reporting general medical practices on COVID-19.

In this study, we propose to conduct a dedicated comprehensive survey on the adoption of machine learning to fight the COVID-19 pandemic from a different perspective, including an extensive literature review and a bibliometric analysis. To the best of our knowledge, this study is the first comprehensive analysis of research output focusing on several possible applications of machine learning techniques for mitigating the worldwide spread of the ongoing COVID-19 pandemic. We are mindful that other publications might not be captured in our scope because the current study is only limited to the eight academic databases mentioned in Table 1. We are also very dependent on the indexing of the databases used, which is akin to any other bibliometric research study.

Table 1 Academic databases.

Academic database	Link	
DBLP	http://dblp.uni-trier.de/	
ACM digital library	http://dl.acm.org/	
IEEEXplore	https://ieeexplore.ieee.org/	
Sciencedirect	http://www.sciencedirect.com/	
Springerlink	https://link.springer.com/	
PubMed	https://pubmed.ncbi.nlm.nih.gov/	
Scopus	https://www.scopus.com/	
Web of science	https://apps.webofknowledge.com/	

Other sections of the study are organized as follows: “Methodology” presents the methodology for the survey. “Theoretical Background of Machine Learning Algorithms” presents the rudiments of the major machine learning algorithms used in fighting the COVID-19 pandemic. “COVID-19 Machine Learning Adoption-Oriented Perspective” presents the adoption of machine learning to fight COVID-19. “COVID-19 Datasets” unravels the different sources of COVID-19 datasets. “Survey and Bibliometric Analysis” discusses the survey and bibliometric analysis. “Challenges and Future Research Opportunities” unveils challenges and future research directions before the conclusion in “Conclusions”. Figure 1 presents the visual structure of the survey article, which is similar to the work in (Mohammadi et al., 2018).

Figure 1 Graphical representation of the survey structure.

Methodology

This section presents the protocol followed to survey the adoption of machine learning in fighting COVID-19. The survey was performed based on a systematic literature review procedure in computer science given by Weidt & Silva (2016). The search keywords, techniques, data sources, databases, and inclusion/exclusion criteria used are discussed.

Search keywords

Initial search keywords were carefully selected based on the defined research goal. After an initial search, multiple keywords were formulated from new words found in several relevant articles. The keywords were later reduced to suit the objective of the research. The keywords used for the study included “deep learning, COVID-19,” “convolutional neural network, COVID-19,” “long short term memory, COVID-19,” “artificial neural network, COVID-19,” “machine learning, COVID-19,” “decision tree, COVID-19,” “COVID-19 diagnosis tool,” and “COVID-19 decision support system.”

Data sources

The defined search keywords were used to retrieve relevant journal articles and conference papers published by prominent peer-reviewed journals indexed in various academic databases. Table 1 shows the different academic databases used to obtain articles for the survey.

Article inclusion/exclusion criteria

Inclusion/exclusion criteria were set up based on the research aim to decide which articles are eligible for the next review stage. Articles that meet the inclusion criteria were considered relevant for the research, and those that do not meet the inclusion criteria were excluded. The set inclusion/exclusion criteria are provided in Table 2.

Table 2 Article inclusion/exclusion criteria.

Inclusion criteria	Exclusion criteria	
The review only focuses on COVID-19	Other viral infections and health issues were not considered relevant in the survey	
Only articles that applied machine learning techniques to fight COVID-19 were considered	Articles using techniques other than machine learning techniques were excluded	
Articles/conference papers published by prominent and indexed journals were included	Articles/conference papers published by non-indexed journals were excluded. The article uploaded as a preprint in preprint servers such as bioRxiv, medRxiv, arXiv, etc. without peer review were excluded	
Only articles written in the English language were considered for inclusion	Articles written in languages other than English were excluded	

Article selection

Article selection for this research followed a three-stage analysis. The first analysis stage considered only the titles and abstracts of the articles to extract relevant articles. The second analysis stage considered the analysis of the abstract, introduction, and conclusion to refine the selection in the first stage. At the third and final analysis stages, articles were read thoroughly, and a threshold was set to rate the quality of articles in terms of their relevance to the research. A article was selected if it reported an empirical application of machine learning to fight COVID-19 similar to Rodriguez-Morales et al. (2020). Articles that met the threshold value were selected, and those below the threshold were dropped. Figure 2 shows the total number of articles obtained from the academic databases and the final number of articles considered for the research after applying all the extraction criteria.

Figure 2 Article selection process.

Bibliometric protocol

VOSviewer software was used to present a bibliometric analysis of the existing literature on COVID-19. VOSviewer software is a tool for constructing and visualizing bibliometric maps of items, such as journals, research, or individual publications. These maps can be created based on citation, bibliographic coupling, co-citation, or co-authorship relations. The bibliometric analysis software also offers text mining functionality that can be used to construct and visualize co-occurrence networks of important terms extracted from a body of scientific literature (see www.vosviewer.com). We only used 1,178 publications with the keyword “novel coronavirus” and 98 publications with the keyword “COVID-19 and artificial intelligence” that were retrieved from Scopus and Web of Science academic databases for the bibliometric analysis presented in this study. Only 57 document results were extracted using the keyword “COVID-19 and machine learning” from the same academic database. Articles from other sources were not considered because most of these publications have not been peer-reviewed and are not available online in the form of preprint publications.

Theoretical Background of Machine Learning Algorithms

Numerous machine learning algorithms exist in the literature. In this section, we discuss the basic operations of the major machine learning algorithms used in fighting the COVID-19 pandemic to provide the readers with a basic concept of how the algorithms operate in achieving their goal, especially novice researchers, thus making the study self-contained. The main algorithms include artificial neural network (ANN), convolutional neural network (CNN), and long short term memory (LSTM).

Artificial neural network

Artificial neural network is a machine learning algorithm designed to emulate the human brain. Similar to neurons in the human brain, ANN consists of interconnected nodes. A given ANN consists of three essential components, namely, node (neuron) character, network topology, and several learning rules (Livingstone, 2008). Signal processing strategy involving the number of inputs and outputs associated with a node, the weight of each input and output, and the activation function are all determined by the node character. The organization of the nodes and the interconnection between them are determined by the network architecture, which usually consists of three underlying layers (input, output, and hidden layers). Learning is employed to train the network. The learning rules decide on weight initialization and adjustment. An ANN is said to have learnt if it can process probabilistic, fuzzy, noisy, and vague data with an insignificant negative effect on the output quality, and generalize acquired knowledge to unknown tasks (Basheer & Hajmeer, 2000). A basic ANN model consists of multiple nodes such that every node receives multiple inputs from other nodes through connections having associated weights. The weighted sum of the inputs is passed through a threshold gate. The node is activated and transmits the output to another node only if the weighted sum of its input exceeds the threshold (Basheer & Hajmeer, 2000).

Convolutional neural network

Convolutional neural network is the most commonly used deep learning algorithm (Haque & Neubert, 2020). It is a discriminative deep learning algorithm formed by a stack of multiple convolutional and pooling layers (Deng, 2014). The major strength of CNN lies in its ability to perform parameter sharing, sparse interaction, and equal representation. The network is faster and easier to train due to its utilization of local connections and sharing of weights (Pouyanfar et al., 2018). A typical CNN receives an image as its input and has neurons arranged in a 3D form connecting to only a portion of the previous layer (Haque & Neubert, 2020). The architecture of CNN is presented in series of stages, where the first stages consist of convolution and pooling layers, and the final stage is composed of a fully connected layer (Lecun, Bengio & Hinton, 2015).

Convolutional neural network receives 3D input x in the form m×m×r, where m is the height and width of the input, and r is the channel number, for example, in RGB r=3. The convolution layer has several filters known as kernels k with a size of n×n×q where n<x and q≤r. The kernels share the same parameters of weight (Wk) and bias (bk). During convolution, k feature maps are generated (hkh), each having size m−n−1. The convolution layer finds the dot product between w and input x, and applies activation function f to the output expressed as (Pouyanfar et al., 2018): hk=f(Wk∗x+bk). The pooling layer prevents overfitting and hastens training by reducing the length of the feature map (Pouyanfar et al., 2018; Zhao et al., 2019). Maximum and average pooling are the widely applied pooling methods. Finally, the fully connected SoftMax layers are deployed for prediction (Zhao et al., 2019).

Long short term memory

The LSTM deep learning network is a type of recurrent neural network that can recall data features over several intervals of time (Liu et al., 2019). It was developed to solve the vanishing gradient weakness of the recurrent neural network (Karim et al., 2018). The hidden layers of LSTM are treated as memory cells, making the network powerful in handling long- and short-term correlation in a time series (Zhao et al., 2017). The earliest version of LSTM contained memory cells, and input and output gates with no forget gate; the forget gate was later deployed to enable continual learning of tasks by resetting the state of the LSTM (Greff et al., 2016). Its updated architecture consists of multiple LSTM units with each unit having an input gate, a forget gate, an output gate, and a memory cell. Sak, Senior & Beaufays (2014) described the underlying architecture of LSTM as consisting of memory blocks in its hidden layer. The memory blocks have memory cells for the storage of the temporal state of the network with additional units, known as gates, to supervise the information flow. A memory cell has an input gate that manages the inflow of input activations to the memory cell and an output gate to manage the outflow of cell activations to the network. The forget gate is incorporated to forget or reset the memory of the cell adaptively. LSTM computes mapping iteratively at a timestamp t=1toT with an input sequence x=(x1,x2,…,xr) and an output sequence y=(y1,y2,…,yr). LSTM is good at addressing complex sequential machine learning problems (Karim et al., 2018). Deep LSTM architectures consist of stacked LSTM layers (Sak, Senior & Beaufays, 2014). LSTMs are strong in handling temporal dependencies in sequences but weak in dealing with long sequence dependencies (Karim et al., 2018).

Covid-19 Machine Learning Adoption-Oriented Perspective

In the fight against COVID-19, different aspects of artificial intelligence (AI) were applied to curtail its adverse effect (Dananjayan & Raj, 2020). The taxonomy in Fig. 3 was created from the project that involved machine learning in fighting COVID-19. The data used in creating the taxonomy were extracted from the articles that applied the machine learning algorithm to fight COVID-19.

Figure 3 Taxonomy of the machine learning algorithms adopted in fighting COVID-19.

COVID-19 diagnostic tools

Currently, the sensitivities for reverse transcription-polymerase chain reaction (RT–PCR)-based viral nucleic acid assay are used as the reference standard method to confirm COVID-19 infection (Corman et al., 2020). However, such a laboratory test is time consuming, and the supply of test kits may be the bottleneck for a rapidly growing suspicious population even for many developed countries such as the USA. More importantly, initial false-negative or weakly positive RT–PCR test results were found in several later-confirmed cases, while highly suspicious computed tomography (CT) imaging features were present (Xu et al., 2020; Xie et al., 2020). The treatment and screening of COVID-19 can be more effective when deep learning approach, CT features, and real-time RT–PCR results are integrated (Li et al., 2020a). AI and deep learning can assist in developing diagnostic tools and deciding on treatment (Rao & Vazquez, 2020; Shi et al., 2020). As a result, many diagnostic tools were developed based on the machine learning algorithm to fight COVID-19. For example, Apostolopoulos & Mpesiana (2020) applied transfer learning with CNN to detect COVID-19 from X-ray images containing common bacterial pneumonia and normal incidents and established COVID-19 infection. Transfer learning CNN was used to diagnose COVID-19 cases from X-ray datasets. The results indicated that VGG19 diagnosed COVID-19 confirmed cases with better accuracy on two- and three-classification problems compared with MobileNet v2, Inception, Xception, and Inception ResNet v2. The proposed approach can help develop a cost-effective, fast, and automatic COVID-19 diagnostic tool, and reduce the exposure of medical workers to COVID-19. Similarly, Rahaman et al. (2020) developed an automated computer-aided diagnosis (CAD) system for the detection of COVID-19 samples from healthy cases and cases with pneumonia using chest X-ray (CXR) images. Their study demonstrated the effectiveness of applying deep transfer learning techniques for the identification of COVID-19 cases using CXR images.

Ardakani et al. (2020) were motivated by the time consumption and high cost of the traditional medical laboratory COVID-19 test to investigate the performance of 10 well-known CNNs in diagnosing COVID-19. The 10 variants of CNN included VGG-16, AlexNet, Xception, VGG-19, ResNet-101, SqueezeNet, ResNet-50, GoogleNet, MobileNet-V2, and ResNet-18. All the CNN variants were applied on CT scan images because the CT slice is a fast method of diagnosing patients with COVID-19. The diagnostic results of the CNN variants indicated that ResNet-101 and Xception outperformed the other CNN variants in diagnosing COVID-19. They concluded that ResNet-101 has a high sensitivity in characterizing and diagnosing COVID-19 infections. Therefore, it can be used as an alternative tool in the department of radiology for diagnosing COVID-19 infection. It is cheaper and faster compared with traditional laboratory analysis.

Butt et al. (2020) applied CNN for the detection of COVID-19 from the chest CT scan of patients. CNN was found very fast and reliable in the detection of COVID-19 from a chest CT scan compared with the conventional RT–PCR testing. In summary, the CNN model is fast and reliable in detecting COVID-19 infection. Huang et al. (2020b) applied a deep learning algorithm on a chest CT scan of a patient with COVID-19 to quantify lung burden changes. The patients with COVID-19 were grouped into mild, moderate, severe, and critical based on findings from the chest CT scan, clinical evaluation, and laboratory results. Deep learning algorithm was applied to assess the lung burden changes. They found that the assessment of lung opacification measured on the chest CT scan substantially differed from that of the clinical groups. The approach can remove the subjectivity in the initial assessment of COVID-19 findings.

Mei et al. (2020) proposed a joint model comprising CNN, support vector machine (SVM), random forest (RF), and multilayer perceptron integrated with chest CT scan result and non-image clinical information to predict COVID-19 infection in a patient. CNN was run on the CT image, while the other algorithms classified COVID-19 using the non-image clinical information. The output of the CNN and the different algorithms were combined to predict the patient’s COVID-19 infection. The diagnostic tool can rapidly detect COVID-19 infection in patients. Liu et al. (2020a) used logistic regression for the prediction of COVID-19 infection sliding to the severity of the COVID-19 cohort. The results of the study showed that the CT quantification for the pneumonia lesions could predict the progression of a patient with COVID-19 to a severe stage at an early, non-invasive level. This situation can provide a prognostic indicator for coronavirus clinical management. Jiang et al. (2020) applied a machine learning algorithm to predict COVID-19 clinical severity. They developed a predictive tool that predicts patients at risk for increased COVID-19 severity at the first presentation. The survey can help in the optimal utilization of scarce resources to cope with the COVID-19 pandemic.

Hurt, Kligerman & Hsiao (2020) collected CXR images from patients with COVID-19 in China and America. They applied a deep learning algorithm for the early diagnosis of COVID-19 from the CXR. They found that deep learning predicted and consistently localized areas of pneumonia. The deep learning algorithm can diagnose a patient’s COVID-19 infection early. Loey, Smarandache & Khalifa (2020) were motivated by the insufficient COVID-19 dataset to propose a generative adversarial network (GAN) and CNN variant to detect COVID-19 in patients. GAN was used to generate more X-ray images. Googlenet, Alexnet, and ResNet18 were applied as the deep transfer learning models. They found that Googlenet and Alexnet scored 80.6%, 85.2%, and 100%, respectively, in the four-, three-, and two-class classification problem, respectively. The study’s method can facilitate the early detection of COVID-19 and reduce the workload of a radiologist. Wu et al. (2020) proposed a multi-view ResNet50 for the screening of COVID-19 from chest CT scan images. ResNet50 was trained with the multi-view chest CT scan images. The results showed that the multi-view ResNet50 fusion achieved a high performance compared with the single view. The diagnosis tool developed can reduce the workload of a radiologist by offering fast, accurate COVID-19 diagnosis. Ucar & Korkmaz (2020) developed a rapid COVID-19 diagnosis tool from X-ray images based on SqeezeNet (a pre-defined CNN) and the Bayesian optimization method. The SqueezeNet hyperparameters were optimized using the Bayesian optimization method. Bayesian optimization-based SqueezeNet was applied to detect COVID-19 from X-ray images labeled normal, pneumonia, and COVID-19. Bayesian-based SqueezeNet outperformed the baseline diagnostic tools.

Toğaçar, Ergen & Cömert (2020) applied CNN for the exploitation of social mimic and CXR based on fuzzy color and the stacking method to diagnose COVID-19. The stacked data were trained using CNN, and the features obtained were processed with mimicking social optimization. The compelling features were used for classification into COVID-19, pneumonia, and standard X-ray imagery using SVM. Singh et al. (2020) used CNN and multi-objective differential evolution (MODE) for the early detection of COVID-19 from a chest CT scan image. The initial parameters of the CNN were tuned using MODE to create a MODE-based CNN and classify patients with COVID-19 based on positive or negative chest CT scan images. MODE-based CNN outperformed the competitive models (ANN, ANFIS, and traditional CNN). The proposed method is beneficial for COVID-19 real-time classification owing to its speed in diagnosing COVID-19. Salman et al. (2020) constructed a CNN-based COVID-19 diagnostic tool for the detection of COVID-19 from CXR images. CNN–inceptionV3 was applied to detect COVID-19 from 130 X-ray images of patients infected with COVID-19 and 130 normal X-ray images. The results indicated that CNN–inceptionV3 could detect COVID-19 from the X-ray images and reduce the testing time required by a radiologist.

Ozturk et al. (2020) used CNN to develop an automated tool for diagnosing COVID-19 from raw CXR images. Binary and multi-class categories were experimented on using a CNN with 17 convolution layers with a different filter on each convolution layer. The model can be used for the early screening of patients with COVID-19 and assist the radiologist in validating COVID-19 screening. Li et al. (2020b) developed an automated framework based on CNN for the detection of COVID-19 from chest CT scan and differentiate it from community-acquired pneumonia. The study collected data from 3,322 patients comprising 4,356 chest CT scans. CNN was applied to detect patients with COVID-19 and typical community pneumonia. The experiment results showed that CNN can distinguish patients with COVID-19 from those with community-acquired pneumonia and other similar lung diseases. The proposed framework automated the COVID-19 testing and reduced the testing time and fatigue. Yang et al. (2020a) applied densely connected convolutional networks optimized with stochastic gradient descent algorithm for the detection of COVID-19 from chest CT scan images. Oh, Park & Ye (2020) applied patch-based CNN–ResNet-18 (P-CNN) due to lack of sufficient training data for diagnosing COVID-19 from CXR images. The study used imaging biomarkers of the CXR radiographs. P-CNN ResNet-18 was applied, and P-CNN produced clinically salient maps that are useful in diagnosing COVID-19 and patient triage. P-CNN ResNet-18 achieved the best result compared with the baselined algorithm performance. The limited amount of data can be used for COVID-19 diagnoses and were interpretable. Table 3 summarizes the diagnostic tools developed based on machine learning. Refer to Dong et al. (2020) for an engaging research review on the role of imaging in the detection and management of COVID-19 disease spread.

Table 3 The summary of the COVID-19 diagnostic tools based on machine learning algorithms.

Reference	Algorithm	Performance	Contribution	Benefit	
Apostolopoulos & Mpesiana (2020)	Transfer Learning with CNN	Accuracy: 98.75%	Device approach for automatic diagnostic of COVID-19 based on X-ray	Cost-effective, fast diagnosis and reduce exposure of medical workers to COVID-19	
Sensitivity: 92.85%	
Specificity: 98.75%	
Ardakani et al. (2020)	Variants of CNN
ResNet 101	Accuracy: 99.51%	Automated characterization and diagnosis of COVID-19 infection.	Cheaper and faster compared to the traditional laboratory analysis of COVID-19. Reduces medical worker’s workload.	
Sensitivity: 100%	
Specificity: 99.02%	
Butt et al. (2020)	CNN	AUC: 0.996	Outperform the traditional RT-PCR testing of COVID-19	fast and reliable in detecting COVID-19 pandemic.	
Sensitivity: 98.2%	
Specificity: 92.2%.	
Huang et al. (2020b)	Deep learning algorithm	Not Applicable as a result of ANOVA analysis	The assessment of the lung opacification measured is significantly different from the clinical groups	The approach has the potential to remove the subjectivity in the initial evaluation of COVID-19 findings as well as follow up pulmonary	
Li et al. (2020b)	CNN	Sensitivity:87%	The automated framework that differentiates COVID-19 from pneumonia	Automated the COVID-19 testing process, reduces the testing time and fatigue.	
Specificity:92%	
AUC: 95%	
Liu et al. (2020a)	P-CNN	Sensitivity: 100%	Diagnose COVID-19 with limited data and present new probabilistic Grad-CAM salient map	Limited amount of data can be used for the COVID-19 diagnoses, and it is interpretable.	
Precision: 76.9%	
Ozturk et al. (2020)	CNN	Sensitivity: 85.35%	Improve efficiency and automate the COVID-19 screening process	Automate the process of COVID-19 diagnoses to reduce fatigue	
Specificity: 92.18%	
Accuracy: 87.02%	
Salman et al. (2020)	CNN	Sensitivity: 100%	Automated COVID-19 screening process	Reduces diagnostic time	
Specificity: 100%	
Accuracy: 100%	
Singh et al. (2020)	MODE based CNN	Sensitivity: > 90%	Diagnose COVID-19 with better accuracy than the competitive models	The proposal is beneficial to COVID-19 real-time classification	
Specificity: > 90%	
Toğaçar, Ergen & Cömert (2020)	CNN-SVM	Overall accuracy: 99.27%	Contributed to the efficient detection of COVID-19	Automated detection of COVID-19 patient	
Ucar & Korkmaz (2020)	Bayesian-based SqueezeNet	Accuracy: 100%	Presents alternative rapid COVID-19 diagnostic tool based on deep Bayes-SqueezeNet	It will be of benefit to healthcare professionals in diagnosing COVID-19 efficiently.	
Specificity: 99.67%	
Wu et al. (2020)	ResNet50	Accuracy: 0.819	Provide COVID-19 diagnosis from multiple view	Reduce the workload of a radiologist by offering fast and accurate COVID-19 diagnosis	
Sensitivity: 0.760	
Specificity: 0.811	
Loey, Smarandache & Khalifa (2020)	Googlenet, Alexnet and RestNet18	Googlenet, Alexnet and Googlenet scored 80.6%, 85.2% and 100% in the 4, 3 and 2 classes classification problems	Generate sufficient COVID-19 data to improve COVID-19 diagnosis	Early detection of COVID-19 and reduce the workload of radiologist	
Hurt, Kligerman & Hsiao (2020)	Deep learning	Not provided	Improve the detection COVID-19 from X-ray	Early detection of COVID-19	
Jiang et al. (2020)	Machine learning algorithm	Accuracy: 70–80%	Detect COVID-19 severity in a patient at the initial presentation	Help in optimal utilization of scarce resources to cope with COVID-19	
Liu et al. (2020a)	Logistic regression	ROC: 0.93	Applied CT quantification of pneumonia to predict progression to COVID-19 severity	provide a prognostic indicator for COVID-19 clinical management	
Confidence interval: 95%	
Mei et al. (2020)	Deep ensemble algorithm	ROC: 0.92	Predict COVID-19 with both image and none image clinical information	The ensemble diagnostic tool can detect COVID-19 patients rapidly	
Accuracy: 68%	
Sensitivity: 84.3%	
Specificity: 82.8%	
Yang et al. (2020a)	Densely connected convolutional networks	Accuracy: 92%	Detect COVID-19 from CT scan via densely connected convolutional networks	Reduce radiologist workload	
Sensitivity: 97%	
Specificity: 87%	

COVID-19 decision support system

The decision support system related to COVID-19 can help decision/policymakers formulate policy to curtail COVID-19. Many COVID-19 decision support systems were developed based on machine learning approaches. For example, Ayyoubzadeh et al. (2020) applied LSTM and linear regression to predict the number of positive cases in Iran. LSTM and linear regression were used on Google search data to predict the COVID-19 cases in Iran. The results indicated that linear regression outperforms LSTM in predicting the positive cases of COVID-19. The algorithm can predict the trend of the COVID-19 pandemic in Iran, which can help policymakers plan the allocation of medical resources. Chimmula & Zhang (2020) applied deep LSTM for forecasting COVID-19 transmission and possible COVID-19 ending period in Canada and other parts of the world. The transmission rate of Canada was compared with that of Italy and the USA. The future outbreak of the COVID-19 pandemic was predicted to help Canadian decision makers monitor the COVID-19 situation and prevent the future transmission of the epidemic in Canada.

Liu et al. (2020b) proposed ANN in modeling the trend of COVID-19 and restoring the operational capability of medical services in China. ANN was used for modeling the pattern of COVID-19 in Wuhan, Beijing, Shanghai, and Guangzhou. Autoregressive Integrated Moving Average (ARIMA) was applied for the estimation of nonlocal hospital demands for the period of COVID-19 pandemic in Beijing, Shanghai, and Guangzhou. The results indicated that the number of people infected with COVID-19 would increase by 45%, while death would increase by 567%. COVID-19 will reach its peak by March 2020 and toward the end of April 2020. This finding will assist policymakers and health officials in planning to deal with challenges of the unmet medical requirement of other diseases during the COVID-19 pandemic.

Pirouz et al. (2020) proposed a group method of data handling in a neural network to predict the number of COVID-19 confirmed cases based on weather conditions. The dominant weather condition used included temperature, city density, humidity, and wind speed. The results indicated that humidity and temperature have a substantial influence on COVID-19 confirmed cases. Temperature and humidity influence COVID-19 negatively and positively, respectively. These results can be used by decision makers to manage the COVID-19 pandemic. Yang et al. (2020b) applied LSTM to predict the COVID-19 trend in China. The prediction model indicated that the COVID-19 pandemic should peak toward the end of February 2020 and start declining at the end of April 2020. The prediction model can be used by authorities in China to decide in controlling the COVID-19 pandemic. Vaid, Cakan & Bhandari (2020) adopted a machine learning approach to predict COVID-19 potential infections based on reported cases in North America. Critical parameters were identified from dimension reduction. Passed diseases were inferred from recent fatalities using a hierarchical Bayesian estimator. The model predicted potential COVID-19 infections in North America. Policymakers in North America can use the projection to curtail the effect of the COVID-19 pandemic. Tuli et al. (2020) developed a machine learning COVID-19 predictive model and deployed it in the cloud computing environment for real-time tracking of COVID-19, predicting the growth and potential thread of COVID-19 in different countries worldwide. Government and citizens can use the results for proactive measures to fight COVID-19.

Tiwari, Kumar & Guleria (2020) used a machine learning approach to predict the COVID-19 pandemic number of cases, recoveries, and deaths in India based on data from China. The prediction results indicated that COVID-19 would peak between the third and fourth week of April 2020. The Indian government can use the study to formulate policies and decide on mitigating the spread of COVID-19. Ribeiro et al. (2020) evaluated six machine learning algorithms, namely, Cubis regression (CUBIST), RF, ridge regression (RIDGE), support vector regression (SVR), ARIMA, and stacking-ensemble learning (SEL), on COVID-19 datasets collected in Brazil to predict confirmed cases for 1, 3, and 6 days ahead. They found that SVR outperformed RIDGE, ARIMA, RF, CUBIST, and SEL. The study can help monitor COVID-19 cases in Brazil and facilitate critical decisions on COVID-19. Tummers et al. (2020) applied k-means to cluster documents based on COVID-19 and people with intellectual disability. Table 4 summarizes the studies on COVID-19 decision support system.

Table 4 Summary of the adoption of machine learning approaches in building COVID-19 decision support systems.

Reference	Algorithm	Performance	Contribution	Benefit	
Ayyoubzadeh et al. (2020)	LSTM & Linear regression	LSTM: RMSE 27.187	Predict COVID-19 positive cases in Iran	The algorithm can predict the trend of the COVID-19 pandemic in Iran, which can help policymakers to plan for the allocation of medical resources	
Linear Regression: RMSE 7.562	
Chimmula & Zhang (2020)	LSTM	RMSE: 34.83	Forecast COVID-19 transmission in Canada	Help decision-makers in monitoring and curtailing future transmission of COVID-19 in Canada	
Accuracy: 92.6%	
Liu et al. (2020b)	ANN	Not Applicable	Estimated the trend of COVID-19 in China	Help policymakers and health officials attend to the need of other diseases during the COVID-19 pandemic	
Ribeiro et al. (2020)	Support vector regression	MAE: 79.17	Provide future COVID-19 confirmed cases in brazil	monitoring COVID-19 cases in Brazil and help decision-makers in taken critical decision about COVID-19	
Tiwari, Kumar & Guleria (2020)	Machine learning algorithm (not specified)	MAE & RSME—graphical	The predicted peak period of COVID-19 in India	Help India policymakers decide on COVID-19 to mitigate its spread	
Tuli et al. (2020)	Machine learning algorithm (not specified)	MSE: 9.32E+06	Provide real life COVID-19 predictions	Government and citizens can use the results for proactive measures to fight COVID-19	
Vaid, Cakan & Bhandari (2020)	Machine learning algorithm (not specified)	Not reported	Predict Potential COVID-19 infections	Policymakers in North America can use the projection to curtail the effect of COVID-19 pandemic	
Yang et al. (2020b)	LSTM	Confidence Interval: 95%	Predict COVID-19 trend in China	Authorities in China to decide to control the COVID-19 pandemic	
Pirouz et al. (2020)	Group method of data handling neural network	Accuracy: 85.7%	Predict COVID-19 pandemic based on weather condition	Help in managing COVID-19 pandemic	

COVID-19 from genome sequences

The protein sequence of COVID-19 can be collected to apply the machine learning approach for the prediction of COVID-19 (Qiang et al., 2020). For example, Qiang et al. (2020) predicted the infection risk of non-human origin of COVID-19 from spike protein for prompt alarm using RF. The genome data comprised of non-human COVID-19 origin (positive) and human COVID-19 origin (negative). RF was applied for the training to predict non-human COVID-19 origin. The results showed that the RF model achieved high accuracy in predicting non-human COVID-19 origin. The study can be used in COVID-19 genome mutation surveillance and exploring evolutionary dynamics in a simple, fast, and large-scale manner. Randhawa et al. (2020) combined decision tree and digital signal processing (DT–DSP) to detect the COVID-19 virus genome and identified the signature of intrinsic COVID-19 viruses’ genome. DT–DSP was applied to explore over 5,000 viral genome sequences with 61.8 million by the 29 viral sequences of COVID-19. The result obtained supported the bat origin of COVID-19 and successfully classified COVID-19 with 100% accuracy as sub-genus Sarbecovirus within Betacoronavirus. DT–DSP is a reliable real-time alternative taxonomic classification. Table 5 summarizes the studies.

Table 5 The summary of detecting COVID-19 from genome sequence via an algorithm.

Reference	Algorithm	Performance	Contribution	Benefit	
Qiang et al. (2020)	Random Forest	Accuracy: 98.18%	Able to detect none human COVID-19 origin from spike protein	used in COVID-19 genome mutation surveillance	
Mathew Correlation Coefficient: 0.9638	
Randhawa et al. (2020)	Decision Tree	Accuracy: 100%	Successfully used intrinsic viral genomic signatures to classify COVID-19 with 100% accuracy	The DT-DSP is a reliable real-time alternative for the classification of taxonomic	

COVID-19 drug discovery

Machine learning and AI provide approaches for the speedy processing of a large amount of collected medical data generated daily as well as the extraction of new information from transversely different applications. In the prediction of disease, a viral mutation can be forecast before the emergence of new strains. It also allows the prediction of new structure and availability of broader structural information. Efficient drug repurposing can be achieved in mining existing data. The stages for the development of COVID-19 drugs are as follows (Park et al., 2020): Disease prediction: The prediction of future-generation viral mutation can be accomplished by AI and machine learning approaches. Structural analysis: The COVID-19 structure and primary functional site are characterized. Drug repurposing: For insight into new disease treatment, existing drug data are mined. New drug development: Efficiencies across the entire pharmaceutical life cycle are achieved by rapid processing. Ke et al. (2020) applied machine learning to identify drugs already marketed that can treat COVID-19. They compiled two independent datasets to develop two machine learning models. The first model was built based on drugs that are known to have antiviral activities. The second model was built based on 3C-like protease inhibitors. The database of market-approved drugs was screened by the machine learning model to predict the drugs with potential antiviral activities. The drugs predicted to have antiviral activities were evaluated against the antiviral activities by a cell-based feline infectious peritonitis virus duplication assay. The assay results were the machine learning model feedbacks for incremental learning of the model. Finally, 80 marketed drugs were identified to have potential antiviral activities. Old drugs with antiviral activities against feline infectious peritonitis COVID-19 were found.

COVID-19 vaccine development

Typically, the immune system is prepared to elicit antibody or cell-mediated responses against a pathogen by a vaccine that protects the body from infectious diseases. Immunogenicity is the vaccine ability to the response. For a long-time, effective immunity, the vaccine has to properly activate innate, adaptive responses (Klein, Jedlicka & Pekosz, 2010).

The following phases should be adopted to develop a COVID-19 vaccine (Gonzalez-Dias et al., 2020): Dataset preparation: The quality of the data to be used influences the machine learning algorithm. Thus, preparing quality data before feeding into the algorithm is sacrosanct. Data come in different sizes ranging from small, medium, and large. Data quality must be ensured because a quality immune response is needed. The reliability of the data needs to be guaranteed by ensuring that the serological assay is well qualified in case it is not validated based on known parameters (linearity, specificity, LLOQ, ruggedness, LLOD, ULOQ, and reproducibility). Vaccines and relevant genes: In vaccinology, the machine learning algorithm is trained to discover the combination of genes and the best vaccines parameters. The data for the training are extracted from omics experiment, which will be used to obtain the required combination. Feature selection is performed to find the best representative of the discriminatory gene signatures. Then, the new vaccines are predicted. The three main feature selection methods are filter, wrapper, and embedded. Machine learning algorithm selection: This task is not a straightforward task because many factors must be considered before selecting the appropriate algorithm for the modeling. The choice of the algorithm depends on the nature of the data, and the options include supervised, unsupervised, and semi-supervised learning. For instance, if the data have no output, then unsupervised learning algorithm, for example, k–the nearest neighbor is the possible candidate algorithm for the modeling but is not guaranteed. Many algorithms have to be tested on solving the same problem before the algorithm that produces the best output is selected. Model testing: The performance of the model is tested. The data are partitioned into training and testing; the former is used for training the algorithm, and the latter is used for evaluating the performance of the model using several performance parameters, for example, MSE, accuracy, and F-measure (Gonzalez-Dias et al., 2020). The application of a machine learning algorithm to sift through trillions of compounds of the vaccine adjuvants can shorten the vaccine development time. The machine learning algorithm can be used for screening compounds for a potential adjuvant candidate for the SARS-CoV-2 vaccine (Ahuja, Reddy & Marques, 2020).

Covid-19 Datasets

Data sources

Ahuja, Reddy & Marques (2020) reported that COVID-19 data are now growing. In this section, we present the sources of COVID-19 data to the machine learning community. Given the novelty of the virus, centralizing the collection of sources of data will help researchers access different types of COVID-19-related data and provide them opportunities to work on a different aspect of COVID-19 that may lead to novel discoveries. Table 6 has five columns, where the first, second, third, fourth, and fifth columns represent the reference, data, owners, source/accessibility, and remarks, respectively. We only present the projects that revealed and fully discussed their data sources.

Table 6 Summary of COVID-19 data sources and accessibility.

Reference	Data	Owners	Source/Accessibility	Remarks	
Apostolopoulos & Mpesiana (2020), Ozturk et al. (2020), Salman et al. (2020), Toğaçar, Ergen & Cömert (2020), Ucar & Korkmaz (2020) and Loey, Smarandache & Khalifa (2020)	X-Ray	Cohen	https://github.com/ieee8023/covid-chestxray-dataset	A project with a collection of X-rays images	
Apostolopoulos & Mpesiana (2020)	X-Ray	Kaggle	https://www.kaggle.com/andrewmvd/convid19-X-rays	Dataset_1: 224 images with Covid-19, 700 images with common bacterial pneumonia, and 504 normal images condition	
Dataset_2: 224 cases of Covid-19, 504 healthy instances, and 714 has both bacterial and viral pneumonia	
Ardakani et al. (2020)	CT scan	Ardakani et al. (2020)	16-MDCT scanner (Alexion, Toshiba Medical System, Japan)	The data contained 1020 CT slices from 108 patients confirmed COVID-19 and 86 patients without COVID-19	
Ayyoubzadeh et al. (2020) and Vaid, Cakan & Bhandari (2020)	Time series	Worldometer	https://www.worldometers.info/coronavirus/	The daily new COVID-19 cases from 02/15, 2020, to 03/18/2020 in Iran. Dataset features: previous day’s search trends, previous day cases, and Output: new cases of the current day	
Barbosa & Fernandes (2020)	Genome	Barbosa & Fernandes (2020)	https://data.mendeley.com/datasets/nvk5bf3m2f/1	The data is chaos game representation of SARS-CoV-2 containing both the raw and processed data with 100 instances of SARS-CoV-2 genome	
Butt et al. (2020)	CT scan	Butt et al. (2020)	Butt et al. (2020)	The data comprised of 618 CT scan samples including 219 from 110 COVID-19 patients	
Chimmula & Zhang (2020)	Time series	Johns Hopkins University and Canadian Health authority	Chimmula & Zhang (2020)	The data is available in time series: date, month and year up to 31/04/20	
Huang et al. (2020b)	CT scan	Huang et al. (2020b)	Huang et al. (2020b)	126 COVID-19 patients that underwent a CT chest scan from 1/1/2020 to 3/2/2020	
Li et al. (2020b)	CT scan	Li et al. (2020b)	Li et al. (2020b)	Data from 3,322 patients comprising of 4356 chest CT scans. Data collection period from 16/08/2016 to 17/02/2020	
Liu et al. (2020a)	Time series	1. Tencent news	Real COVID-19 reporting: https://news.qq.com/zt2020/page/feiyan.htm?ADTAG=area	1. COVID-19 time-series data including locations confirmed cases, deaths, recovery cases, and new diagnosed cases	
2. Baidu	
Baidu: http://qianxi.baidu.com	
2. Baidu is an open-source for big data project that provides visualization of population migration	
Oh, Park & Ye (2020)	X-ray	Society of Radiological Technology	Oh, Park & Ye (2020)	Segmentation network dataset with a total of 247 posteroanterior chest radiographs	
Qiang et al. (2020)	Protein sequence	China national genomic data center	https://bigd.big.ac.cn/ncov	The database has an extensive protein sequence	
Randhawa et al. (2020)	Genome	National Center for Biotechnology Information	https://sourceforge.net/projects/mldsp-gui/files/COVID19Dataset	The database has genome dataset with thousands of bp	
Ribeiro et al. (2020)	Time series	Ribeiro et al. (2020)	https://brasil.io/api/dataset/covid19/caso/data/?place_type=state	Daily COVID-19 reports on confirmed cases from different states in Brazil	
Singh et al. (2020)	CT scan	Singh et al. (2020)	Singh et al. (2020)	Chest CT scan images from ICU	
Tiwari, Kumar & Guleria (2020)	Time series	Center for Systems Science and Engineering (CSSE) at Johns Hopkins University (JHU).	https://www.kaggle.com/sudalairajkumar/novel-corona-virus-2019-dataset	Confirmed COVID-19 cases, recovered cases, and death cases from China	
Tuli et al. (2020)	Time series	Hannah Ritchie	https://ourworldindata.org/coronavirus-source-data	The dataset is the Our World in Data provided by Hannah Ritchie	
Wu et al. (2020)	CT scan	Wu et al. (2020)	Wu et al. (2020)	The CT scan images of 495 patients were collected from three hospitals in China	
Yang et al. (2020b)	Time series	1. National health commission of China	1. http://www.nhc.gov.cn/xcs/yqtb/list_gzbd.shtml	SARS and COVID-19 datasets	
2. https://qianxi.baidu.com/	
2. Baidu	
3. http://news.sohu.com/57/26/subject206252657.shtml	
3. SOHU	
Hurt, Kligerman & Hsiao (2020)	X-ray	Hurt, Kligerman & Hsiao (2020)	Hurt, Kligerman & Hsiao (2020)	Chest X-ray images from China and America	
Jiang et al. (2020)	CT scan & clinical characteristics	Jiang et al. (2020)	Jiang et al. (2020)	CT scan images and clinical characteristics	
Liu et al. (2020a)	CT scan	Liu et al. (2020a)	Liu et al. (2020a)	Chest CT scan that was performed using a 64-slice CT scanner without contrast agents.	
Mei et al. (2020)	CT scan and clinical information	Mei et al. (2020)	Mei et al. (2020)	Both clinical information and chest CT scan images were collected for the study	
Pirouz et al. (2020)	Time series	Pirouz et al. (2020)	Pirouz et al. (2020)	Environmental and urban data	
Yang et al. (2020a)	CT scan	Yang et al. (2020a)	Yang et al. (2020a)	CT scan images from a hospital in china	

COVID-19 on CXR and CT scan images

In this section, we discuss the diagnosis of COVID-19 based on X-ray and CT scan images because of their high value in COVID-19 screening. Table 6 shows that researchers heavily utilize X-rays and CT scans in developing a machine-learning-based COVID-19 diagnosis tool. Guan et al. (2020) and Wong et al. (2020) found that portable chest radiography (CXR) has a sensitivity of 59% for the initial detection of COVID-19-related abnormalities. Radiographic abnormalities, when present, mirror those of CT, with a bilateral lower zone, a peripherally predominant consolidation, and hazy opacities (World Health Organization, 2020). The radiological findings of COVID-19 on CXR are those of atypical pneumonia or organizing pneumonia (Kooraki et al., 2020). Although chest CT scans are reportedly less sensitive than CXRs, chest radiography remains the first-line imaging modality of choice used for patients with suspected COVID-19 infection (Hui et al., 2020) because it is cheap and readily available, and can easily be cleaned. For ease of decontamination, the use of portable radiography units is preferred. Chest radiographs are often normal in early or mild disease. According to a recent study of patients with COVID-19 requiring hospitalization, 69% had an abnormal chest radiograph at the initial time of admission, and 80% had radiographic abnormalities sometime during hospitalization. The findings are reported to be most extensive about 10–12 days after symptom onset. The most frequent radiographic findings are airspace opacities, whether described as consolidation or less commonly, ground-glass opacity (GGO) (Wong et al., 2020). The distribution is most often bilateral, peripheral, and lower zone predominant (Rodrigues et al., 2020). Unlike parenchymal abnormalities, pleural effusion is rare (3%) (Wong et al., 2020). According to the Center for Disease Control (CDC), even if a chest CT or X-ray suggests COVID-19, viral testing is the only specific method for diagnosis. Radiography’s sensitivity was reported at only 25% for detection of lung opacities related to COVID-19, among 20 patients seen in South Korea with a reported specificity of 90% (Wen et al., 2020). The X-ray image should be considered a useful tool for detecting COVID-19 which is challenging the healthcare system due to the overflow of patients. As the COVID-19 pandemic grinds on, clinicians on the front lines may increasingly turn to radiography (Casey, 2020). The most frequent findings are airspace opacities, whether described as consolidation or less commonly, GGO. The distribution is most often bilateral, peripheral, and lower zone predominant (Wong et al., 2020). Much of the imaging focus is on CT. In February 2020, Chinese studies revealed that chest CT achieved a higher sensitivity for the diagnosis of COVID-19 compared with initial RT–PCR tests of pharyngeal swab samples (Ai et al., 2020; Fang et al., 2020). Subsequently, the National Health Commission of China briefly accepted chest CT findings of viral pneumonia as a diagnostic tool for detecting COVID-19 infection (Yuen et al., 2020; Zu et al., 2020).

The typical appearance of COVID-19 on chest CT consists of multi-lobar, bilateral, predominantly lower lung zone, rounded GGOs, with or without consolidation, in a mostly peripheral distribution. However, such findings are nonspecific; the differential diagnosis includes organizing pneumonia and other infections, drug reactions, and other inflammatory processes. Consequently, using CT to screen for COVID-19 may result in false positives. Moreover, the presence of abnormalities not typically associated with COVID-19 infection, including pure consolidation, cavitation, thoracic lymphadenopathy, and nodules suggests a different etiology (Bernheim et al., 2020). COVID-19-related chest CT abnormalities are more likely to appear after symptom onset, but they may also precede clinical symptoms. In a retrospective study by Bernheim et al. (2020), 44% of patients presenting within 2 days of symptom onset had an abnormal chest CT, while 91% presenting within 3–5 days and 96% presenting after 6 days had abnormal chest CTs. Shi et al. (2020) found GGOs in 14 of 15 asymptomatic healthcare workers with confirmed COVID-19. Similarly, 54% of 82 asymptomatic passengers with COVID-19 on the Diamond Princess cruise ship had findings of viral pneumonia on the CT (Inui et al., 2020).

In a prospective study by Wang et al. (2020b) pure GGOs were the only abnormalities seen prior to symptom onset. Subsequently, 28% of patients developed superimposed septal thickening 6–11 days after symptom onset. Architectural distortion evolving from GGOs appeared later in the disease course, likely reflecting organizing pneumonia and early fibrosis. Long-term follow-up imaging is also needed to determine the sequelae of SARS-CoV-2 infection. In a retrospective study by Das et al. (2017), 33% of patients who recovered from MERS-CoV developed pulmonary fibrosis; a similar outcome following COVID-19 is likely. Lung ultrasound offers a low-cost, point-of-care evaluation of the lung parenchyma without ionizing radiation. The modality is especially useful in resource-limited settings (Stewart et al., 2020). Peng, Wang & Zhang (2020) found that sonographic findings in patients with COVID-19 correlated with typical CT abnormalities. The predominantly peripheral distribution of lung involvement facilitated sonographic visibility. Characteristic findings included thickened, irregular pleural lines, B lines (edema), and eventual appearance of A lines (air) during recovery. Peng, Wang & Zhang (2020) suggested that ultrasound may be useful in recruitment maneuver monitoring and guide prone positioning.

Chest CT is an indispensable tool for the early screening and diagnosis of suspected COVID-19 infection in patients. Previous studies confirmed that the majority of patients infected with COVID-19 exhibited common chest CT characteristics, including GGOs and consolidation, which reflect lesions affecting multiple lobes or infections in the bilateral lung parenchyma. Increasing evidence suggests that these chest CT characteristics can be used to screen suspected patients and serve as a diagnostic tool for COVID-19-caused acute respiratory diseases (ARDS) (Xu et al., 2020). These findings have led to the modification of the diagnosis and treatment protocols of SARS-CoV-2-caused pneumonia to include patients with characteristic pneumonia features on chest CT but negative RT–PCR results in severe epidemic areas such as Wuhan City and Hubei Province (Liu et al., 2020a). Patients with negative RT–PCR but positive CT findings should be isolated or quarantined to prevent clustered or wide-spread infections.

The critical role of CT in the early detection and diagnosis of COVID-19 becomes more publicly acceptable. However, several studies also reported that a proportion of RT–PCR-positive patients, including several severe cases, had initially normal CXR or CT findings (Fang et al., 2020). According to the diagnostic criteria of COVID-19, patients might have no or atypical radiological manifestations even at the mild or moderate stages because several lesions are easily missed in the low-density resolution of CXR, suggesting that chest CT may be a better modality with a lower false-negative rate. Another possible explanation is that in several patients, the targeted organ of COVID-19 may not be the lung. Multiple-organ dysfunctions, including ARDS, acute cardiac injury, hepatic injury, and kidney injury, have been reported during COVID-19 infection (Huang et al., 2020a). Studies also reported the chest CT appearances in patients with COVID-19 after treatment, suggesting its critical role in treatment evaluation and follow up. For example, a study investigated the change in chest CT findings associated with COVID-19 at different time points during the infection course (Pan et al., 2020). The results showed that most apparent abnormalities on the chest CT were still observable for 10 days but disappeared at 14 days after the initial onset of symptoms. Unexpectedly, a case report showed pre- and post-treatment chest CT findings of a 46-year-old woman whose RT–PCR result became negative, while pulmonary lesions were reversal (Duan & Qin, 2020).

Deep learning applications for COVID-19

Singh et al. (2020) developed a deep CNN, which was applied in the automated diagnosis and analysis of COVID-19 in infected patients to save the time and energy of medical professionals. They tuned and used the hyperparameters of CNN by using multi-objective adaptive differential evolution (MADE). Further in the course of their experiments which were extensively carried out, they used several benchmark COVID-19 datasets. The data used to evaluate the performance of their proposed model were divided into training and testing datasets. The training sets were used to build the COVID-19 classification model. Then, the hyperparameters of the CNN model were optimized on the training sets by using the MADE-based optimization approach. The results from the comparative analysis showed that their proposed method outperformed existing machine learning models such as CNN, GA-based CNN, and PSO-based CNN in terms of different metrics (including F-measure, sensitivity, specificity, and Kappa statistics).

Jaiswal et al. (2020) applied deep learning models for the diagnosis and detection of COVID-19, and it was called DenseNet201-based deep transfer learning (DTL). The authors used these pre-trained deep learning architecture as automation tools to detect and diagnose COVID-19 in chest CT scans. The DTL model was used to classify patients as COVID-19 positive (+ve) or COVID-19 negative (−ve). The proposed model was also utilized to extract several features by adopting its own learned weights on the ImageNet dataset along with a convolutional neural structure. Extensive analysis of the experiments showed that the proposed DTL-based COVID-19 model was superior to competing methods. The proposed DenseNet201 model achieved a 97% accuracy compared with other models and could serve as an alternative to other COVID-19 testing kits.

Li et al. (2020c) developed a fully automated AI system to assess the severity of COVID-19 and its progression quantitatively using thick-section chest CT images. The AI system was implemented to partition and quantify the COVID-19-infected lung regions on thick-section chest CT images automatically. The data generated from the automatically segmented lung abnormalities were compared with those of the manually segmented abnormalities of two professional radiologists by using the Dice coefficient on a randomly selected subset of 30 CT scans. During manual and automatic comparisons, two biomarker images were automatically computed, namely, the portion of infection (POI) and the average infection HU (iHU), which were then used to assess the severity and progression of the viral disease. The performance of the assessments was then compared with patients’ status of diagnosis reports, and key phrases were extracted from the radiology reports using the area under the receiver’s operating characteristic curve (AUC) and Cohen’s kappa statistics. Further in their study, the POI was the only computed imaging biomarker that was effective enough to show high sensitivity and specificity for differentiating the groups with severe COVID-19 and groups with non-severe COVID-19. The iHU reflected the progress rate of the infection but was affected by several irrelevant factors such as the construction slice thickness and the respiration status. The results of the analysis revealed that the proposed deep-learning-based AI system accurately quantified the COVID-19 strains associated with the lung abnormalities, and assessed the virus’ severity and its corresponding progression. Their results also showed that the deep learning-based tool can help cardiologists in the diagnosis and follow-up treatment for patients with COVID-19 based on the CT scans.

Singh, Kumar & Kaur (2020) used a CNN to classify patients with COVID-19 as COVID-19 +ve or COVID-19 −ve. The initial parameters of CNN were tuned by using MODE. The authors adopted the mutation, crossover, and selection operations of the differential evolution (DE) algorithm. They extracted the chest CT dataset of COVID-19-infected patients and decomposed them into training and testing groups. The proposed MODE-based CNN and competitive classification models were then applied to the training dataset. They compared the competitive and proposed classification models by considering different fractions of the training and testing datasets. The extensive analysis showed that the proposed model classified the chest CT images at reasonable accuracy rates compared with other competing models, such as ANN, ANFIS, and CNS. The proposed model was also useful for COVID-19 disease classification from chest CT images.

Asif & Wenhui (2020) implemented a model that automatically detected COVID-19 pneumonia in patients using digital CXR images while maximizing the accuracy in detection by using deep convolutional neural networks (DCNN). Their model named DCNN-based model Inception V3 with transfer learning detected COVID-19 infection in patients using CXR radiographs. The proposed DCNN also provided insights on how deep transfer learning methods were used for the early detection of the disease. The experimental results showed that the proposed DCNN model achieved high accuracy. The proposed model also exhibited excellent performance in classifying COVID-19 pneumonia by effectively training itself from a comparatively lower collection of images.

Hu et al. (2020) implemented a weak supervised deep learning model for detecting and classifying COVID-19 infection from CT images. The proposed model minimized the requirements of manual labeling of CT images and accurately detected the viral disease. The model could distinguish positive COVID-19 cases from non-positive COVID-19 cases by using COVID-19 samples from retrospectively extracted CT images from multiple scanners and centers. The proposed method accurately pinpointed the exact position of the lesions (inflammations) caused by the viral COVID-19 and potentially provided advice on the patient’s severity to guide the disease triage and treatment. The experimental results indicated that the proposed model achieved high accuracy, precision, and classification as well as good qualitative visualization for the lesion detections.

Ayyoubzadeh et al. (2020) conducted a study to predict the incidence and occurrence of COVID-19 in Iran. The authors obtained data from the Google Trends website (recommender systems) and used linear regression and LSTM models to estimate the number of positive COVID-19 cases from the extracted data. Root mean square error and 10-fold cross-validation were used as performance metrics. The predictions obtained from the Google Trend’s website were not very precise but could be used to build a base for accurate models for more aggregated data. Their study showed that the population (Iranians) focused on the usage of hand sanitizer and handwashing practices with antiseptic as preventive measures against the disease. The authors used specific keywords related to COVID-19 to extract Google search frequencies and used the extracted data to predict the degree of COVID-19 epidemiology in Iran. They suggested future research direction using other data sources such as social media information, people’s contact with the special call center for COVID-19, mass media, environmental and climate factors, and screening registries.

Randhawa et al. (2020) integrated supervised machine learning with digital signal processing called MLDSP for genome analyses, which were then augmented by a DT approach to the machine learning component, and a Spearman’s rank correlation coefficient analysis for result validation. The authors identified an intrinsic COVID-19 virus genome signature and used it together with a machine-learning-based alignment-free approach for an ultra-fast, scalable, and highly accurate classification of the COVID-19 genomes. They also demonstrated how machine learning used intrinsic genomic signature to provide a rapid alignment-free taxonomic classification of novel pathogens. The model accurately classified the COVID-19 virus without having a priori knowledge by simultaneous processing of the geometric space of all relevant viral genomes. Their result analysis supported the hypothesis of a bat origin and classified the COVID-19 virus as Sarbecovirus within Betacoronavirus. Also, their results were obtained through a comprehensive analysis of over 5,000 unique viral sequences through an alignment-free analysis of their 2D genomic signatures, combined with a DT use of supervised machine learning, and confirmed by Spearman’s rank correlation coefficient analyses.

Farhat, Sakr & Kilany (2020) reviewed the developments of deep learning applications in medical image analysis which targeted pulmonary imaging and provided insights into contributions to COVID-19. The study covered a survey of various contributions from diverse fields for about 3 years and highlighted various deep learning tasks such as classification, segmentation, and detection as well as different pulmonary pathologies such as airway diseases, lung cancer, COVID-19, and other infections. The study summarized and discussed current state-of-the-art approaches in the research domain, highlighting the challenges, especially given the current situation of COVID-19. First, the authors provided an overview of several medical image modalities, deep learning, and surveys on deep learning in medical imaging, in addition to available datasets for pulmonary medical images. Second, they provided a summarized survey on deep-learning-based applications and methods on pulmonary medical images. Third, they described the COVID-19 disease and related medical imaging concerns, summarized reviews on deep learning application to COVID-19 medical imaging analysis, and listed and described contributions to this domain. Finally, they discussed the challenges experienced in the research domain and made suggestions for future research.

Survey and Bibliometric Analysis

Survey analysis

In this survey, we review the projects that used machine learning to fight COVID-19 from a different perspective. We only considered published papers in reputable journals, and conferences, and no preprint papers uploaded in preprint server was used in the survey. We apprized 30 studies that reported the description of the machine learning approach to fighting COVID-19. We found that machine learning has made an inroad into fighting COVID-19 from a different aspect with potential for real-life applications to curtail the negative effect of COVID-19. Machine learning algorithms such as CNN, LSTM, and ANN that are utilized in fighting COVID-19 mostly reported excellent performance compared with the baseline approaches. Many of the studies complained about the scarcity of sufficient data to carry out large-scale study because of the novelty of the COVID-19 pandemic.

We found that various studies used different COVID-19 data. Figure 4 depicts the type of data used in different studies that applied machine learning algorithm to develop different models for fighting COVID-19 pandemic. The data used to plot Fig. 4 were extracted from machine learning research on COVID-19 (refer to Table 6). The longest bars show that X-rays and CT scans have the highest patronage from the studies. Many of the studies used deep learning algorithms, for example, CNN and LSTM, for the diagnosis of COVID-19 on X-rays and CT scans. The evaluation indicated the excellent performance of the algorithms in detecting COVID-19 on X-rays and CT scan images.

Figure 4 Visual representation of COVID-19 data extracted from different projects.

The CT scan has a great value in the screening, diagnosis, and follow up of patients with COVID-19. The CT scan has now been added as a criterion for diagnosing of COVID-19 (Liu et al., 2020a; Li et al., 2020c). The X-rays with COVID-19 pandemic data project by Cohen hosted on GitHub is receiving unprecedented attention from the research community for accessing freely available data. Figure 5 presents the frequency of machine learning algorithms adopted to fight COVID-19. The longest bar indicates that CNN received the most considerable attention from the researchers working in this domain to fight COVID-19. The likely reason why CNN has the highest number of applications is that most of the data used in detecting COVID-19 infection in patients are images (see Fig. 4). CNN is well known for its robustness, effectiveness, and efficiency in image processing compared with other conventional machine learning algorithms because of its automated feature engineering and high performance. The CNN variant suitable for the diagnosis of COVID-19 from X-ray and CT scan images is ResNet. However, many of the studies did not provide the specific type of CNN adopted for the diagnosis of the COVID-19 from X-ray and CT scan images (see Table 3).

Figure 5 Machine learning algorithms adopted in fighting COVID-19.

Figure 6 shows the different aspects where machine learning algorithms were applied in fighting COVID-19. We found that the studies mainly adopted machine learning algorithms in developing COVID-19 diagnosis tools, decision support system, drug development, and detection from protein sequence. The most extended portion of the pie chart indicates that diagnostic tools attracted the most considerable attention, showing the quest for diagnostic tools in the fight against COVID-19 pandemic because the match starts with a diagnosis before the appropriate treatment is administered to save a life, and incorrect diagnosis can lead to inappropriate medication, resulting in further health complications. Most of the studies that adopted machine learning to develop diagnostic tools intended to reduce the workload of radiologists, improve the speed of diagnosis, automate the COVID-19 diagnostic process, reduce the cost compared with traditional laboratory tests, and help healthcare workers in making critical decisions.

Figure 6 Different aspect of machine learning applications in fighting COVID-19 pandemic.

The studies argued that the diagnostic tool could reduce the exposure of healthcare workers to patients with COVID-19, thus decreasing the risk of spreading COVID-19 to healthcare workers. The second part of the pie chart with the most substantial portion is the decision support system for detecting the rate of spread of the virus, confirmed cases, mortalities, and recovered cases. This information from the decision support system can help the government functionaries, policymakers, decision makers, and other stakeholders in formulating policy that can help fight COVID-19 pandemic.

Figure 7 shows that the publications on the adoption of machine learning to fight COVID-19 started appearing in 2020 with no publications in 2019, although COVID-19 started appearing in late 2019. This situation is likely because a new virus-like the COVID-19 pandemic at the initial stage lacks data and has scant information and uncertainties.

Figure 7 Trend of publications on machine learning applications in fighting COVID-19.

Bibliometric analysis

The primary purpose of conducting a bibliometric analysis study in this study is to reflect the trend of rapidly emerging topics on COVID-19 research, where substantial research activity has already begun extensively during the early stage of the outbreak. Another significance of the bibliometric analysis method presented is to aid in the mapping of research situation on coronavirus disease as reported in several scientific works of literature by the research community. In this section, we present the bibliographic coupling among different article items on machine learning to fight COVID-19. The link between the items on the constructed map corresponds to the weight between them either in terms of the number of publications, common references, or co-citations. These items may belong to a group or a cluster. In the visualization, items within the same cluster are marked with the same color, and colors indicate the cluster to which a journal was assigned by the clustering technique implemented by the VOSviewer software. The circular node may represent the items, and their sizes may vary depending on the weight of the article.

Prolific authors

The bibliographic coupling between the top 25 authors is shown in Fig. 8. The two clusters, namely, red and green, correspond to all authors working on similar research fields “COVID-19” and citing the same source in their reference listings. The similarity in cluster color for the authors also implies that the degree of overlap between the reference lists of publications of these authors is higher. Figure 8 shows the visible names, and other names may not be represented in the constructed map.

Figure 8 Bibliographic coupling among the authors.

Two clusters, namely red (left) and green (right), correspond to all authors working on similar research fields “COVID-19” and citing the same sources in their reference listings.

Productive countries

Figure 9 shows the bibliographic coupling of the topmost productive countries. Here, bibliographic coupling indicates that a common reference list in the articles published by these countries. The five clusters are represented by six colors. Red represents China and the USA with the highest strength in terms of contributions, after which comes India and Iran as the next countries within the red node. Green represents Hong Kong, which appears to have the highest strength, whereas blue is for the United Kingdom and Saudi Arabia that have the highest strength. Yellow denotes Japan, Singapore, Thailand, and Taiwan as the highest contributors. Purple refers to Italy and Canada as the two contributing countries. The link between the red and green clusters are thicker compared with that between the blue and red clusters, or between the blue and purple clusters. The thickness of the link simply depicts the degree of intersection of the literature work between the different locations or countries.

Figure 9 Bibliographic coupling among the countries.

Red represents China and the USA with the highest strength in terms of contributions, after which comes India and Iran as the next countries within the red node. Green represents Hong Kong, which appears to have the highest strength, whereas blue is for the United Kingdom and Saudi Arabia that have the highest strength. Yellow denotes Japan, Singapore, Thailand, and Taiwan as the highest contributors. Purple refers to Italy and Canada as the two contributing countries.

Collaborating institution

Figure 10 shows the bibliographic coupling of the network of collaborating institutions that are affiliated with at least four documents on COVID-19 research. Map construction and analysis show that of the 949 institutions, 28 meet the thresholds of collaborating institutions, including the Guangzhou Center for Disease Control (or Chinese Center for Disease Control and Prevention), Huazhong University of Science and Technology, Wuhan University, Capital Medical University, Chinese Academy of Medical Sciences, University of Hong Kong, Sun Yat-Sen University, and Fudan University. Two clusters are identified by red and green colors. Institutions that fall under the same groups appear to have similar literature background or worked on related research fields.

Figure 10 Bibliographic coupling among institutions.

Two clusters, namely red (left) and green (right), correspond to all authors working on similar research fields “COVID-19” and citing the same sources in their reference listings.

Journals

Bibliographic coupling between journals implies that the papers published in these journals have more common reference lists. Three clusters are depicted on the map with red, blue, and green colors. The links with the highest strength occur between Emerging Microbes Journal, Journal of Virology, and Journal of Infection. This link is closely followed by the links between Eurosurveillance and Journal of Infection, Archive of Academic Emergency Medicine, Chinese Medical Journal, and The Lancet. The Journal of Infection Control and Hospital and Journal of Hospital Infection form the weakest networks of a cluster. Figure 11 illustrates the bibliographic coupling between the considered journals.

Figure 11 Bibliographic coupling among the journals.

Three clusters are depicted on the map with red (left), blue (bottom-centre), and green (right) colors. Each cluster shows COVID-19 published papers with more common reference lists among the associated journals.

Co-authorship and authors

Figure 12 illustrates the co-authors and author map visualization. This analysis aims to produce the visualization of all the major authors publishing together or working on similar research fields. The analysis type is co-authorship, and the unit of analysis is authors. The threshold of the minimum number of articles by an author is 25. Network construction and analysis shows that of 2,381 authors, only nine authors meet the limits. However, the most extensive set of connected entities consists of only eight authors, whose visual representation is depicted in Fig. 12, where only one cluster is denoted by red color. The connected link illustrates that these authors have collaborated on the same project or worked on the same research with a similar focus. The thickness of the link between these three authors indicates more common publications.

Figure 12 Co-authorship and authors’ analysis.

The four main clusters, namely, blue (right), red (bottom-centre), green (top-centre), pink (left) match all the major co-authors and authors publishing together or working on similar research fields.

Impact analysis

Figure 13 illustrates the citation analysis among authors’ institutions. Six clusters are represented using different colors. The red cluster has the highest number of author citations from two institutions, namely, the Huazhong University of Science and Technology, Wuhan University (State Key Laboratory of Virology), and the Department of Microbiology, University of Hong Kong. Figure 14 shows the bibliometric analyses of author citations by journal sources. A link between two journal sources indicates the citation connectivity between the two sources.

Figure 13 Author citation by institution.

Six clusters are represented using different colors to denotes authors citation counts per institution, for which only the red (bottom-left) cluster has the highest number of author citations from two institutions in China.

Figure 14 Author citation by journal source.

Three major clusters, namely, red (left), lemon-green (top-right), and green (bottom-left) were identified in the analysis, as the top-cited journal sources as per author publications.

The connected links between the Journal of Virology and the New England Journal of Medicine in Fig. 14 reveal that a publication from the Journal of Virology has cited another publication that is published in the New England Journal of Medicine or vice versa. The thickness and link strength signify more numbers of citation among the clusters. Therefore, among the different clusters identified in the analysis, the Journal of Virology is the top-cited source by publication from other journal sources.

Challenges and Future Research Opportunities

In this section, we present challenges and future research prospects. More so, Fig. 15 describes the course of conducting the literature survey and opportunities for future research with the possibility of solving the challenges to help expert researchers easily identify areas that need development. The challenges and future research opportunities are presented as follows:

Figure 15 Visual representation of the challenges.

Lack of sufficient COVID-19 data: The primary concern with the research in COVID-19 is the barrier prompted by the lack of adequate COVID-19 clinical data (Alimadadi et al., 2020; Mei et al., 2020; Ayyoubzadeh et al., 2020; Fong et al., 2020; Oh, Park & Ye, 2020; Toğaçar, Ergen & Cömert, 2020; Ucar & Korkmaz, 2020; Belfiore et al., 2020). However, an in-depth analysis of patients with COVID-19 requires much more data (Apostolopoulos & Mpesiana, 2020). Data is the key component in machine learning. Machine learning approaches typically experience a limitation in their efficiency and effectiveness in solving machine learning problems without sufficient data. Therefore, insufficient COVID-19 clinical data can limit the performance of specific machine learning algorithms, such as deep learning algorithms that require large-scale data. In this case, developing machine-learning-based COVID-19 diagnostic and prognosis tools, and therapeutic approaches to curtail COVID-19, and predicting a future pandemic can face a severe challenge in terms of performance due to insufficient COVID-19 clinical data. Alimadadi et al. (2020) suggested global collaborations among stakeholders to build COVID-19 clinical database and mitigate the issue of inadequate COVID-19 clinical data. Existing biobanks containing the data of patients with COVID-19 are integrated with COVID-19 clinical data. We suggest that researchers use GAN to generate additional X-rays and CT scan images for COVID-19 to obtain sufficient data for building COVID-19 diagnosis tools. For example, Loey, Smarandache & Khalifa (2020) were motivated by insufficient data and used GAN to generate more X-ray images and develop a COVID-19 diagnostic tool.

Visualization from X-rays and CT Scan: Figure 4 shows that X-ray and CT scan are the two primary clinical data for detecting COVID-19 infection in patients. Distinguishing patients with COVID-19 and mild symptoms from pneumonia on X-ray images could be visualized inaccurately or cannot be visualized totally (Apostolopoulos & Mpesiana, 2020). We suggest that researchers propose machine learning strategies that can accurately differentiate patients with COVID-19 and mild symptoms from patients with pneumonia symptoms based on X-ray images. COVID-19 that is caused by coronavirus might have a CT scan image characteristic similar to other pneumonia caused by a different virus. In the future, the performance of CNN should be evaluated in classifying COVID-19 and viral pneumonia with RT–PCR (Li et al., 2020b).

Uncertainties: When a new pandemic breaks out, it comes with limited information and very high uncertainly, unlike the commonly known influenza. Therefore, knowledge regarding the new epidemic is not sufficient due to the absence of a prior case that is the same as the recent pandemic. In the case of COVID-19, many of the decision makers relied on SARS for reference because of the similarity, even though it is considerably different from COVID-19. The new pandemic typically poses a challenge to data analytics, considering its limited information and geographical and temporal evolving of the recent epidemic. Therefore, an accurate model for predicting the future behavior of a pandemic becomes challenging due to uncertainty (Fong et al., 2020). We suggest that researchers propose a new pandemic forecasting model based on active learning in machine learning to reduce the level of uncertainty, typically accompanying new pandemics such as COVID-19.

Non-uniform pattern: Liu et al. (2020a) applied Susceptible, Exposed, Infectious, Recovered (SEIR) for modeling COVID-19. However, the SEIR model could not capture the complete number of infected cases, while the study ignored imported COVID-19 confirmed cases. SEIR was based on the people’s natural distribution and cannot apply to welfare institute an example of different population distribution. The epidemiological trend of COVID-19 was not predicted accurately by the SEIR model under the viral mutation and specific ant-viral therapy development scenario. The SEIR model was unable to simulate non-uniform patterns, such as the issue of increasing medical professionals and bed capacity (Liu et al., 2020a). We suggest that researchers propose a machine-learning-based strategy for handling the non-uniform pattern in the future and consider all the other factors not considered in the study.

Insufficient regional data: Adequate COVID-19 data for a particular region are lacking because the capacity to gather reliable data is not uniform across regions worldwide. This situation can bring a challenge to the region without available COVID-19 data. We suggest that researchers apply the cross-population train–test model because a model trained in a different region can be used to detect COVID-19 in a different region. For example, the model trained to detect the new virus in Wuhan, China, can be used in Italy (Santosh, 2020).

Image resolution: The resolution of the X-ray images affects the performance of the machine learning algorithm. Dealing with low-resolution images typically poses a challenge to the machine learning approach. Variable size of the resolution dimension has a negative effect. Successful performance cannot be achieved if the input images of the data have different sizes. The original image resolution dimension, structured images, and stacking technique need to be the same (Toğaçar, Ergen & Cömert, 2020). We suggest high-resolution X-ray images for developing COVID-19 diagnostic and prognosis system with the ability to work with low-resolution X-ray images.

Outliers and noise: At the early phase of COVID-19, the COVID-19 data contained many outliers and much noise (Tuli et al., 2020). An outlier in data is a subset of the data that appears with inconsistencies from the remaining data. Outliers typically lower the fit of a statistical model (Bellazzi, Guglielmann & Ironi, 1998). The presence of outliers and noise in COVID-19 makes predicting the correct number of COVID-19 cases challenging (Tuli et al., 2020). Dealing with outliers and noise in data increases data engineering efforts and expenses. We suggest that researchers propose a robust machine learning approach that can effectively handle outliers and noise in COVID-19 data.

Transparency and interpretability: The limitation of deep learning algorithms is a deficiency in terms of transparency and interpretability. For instance, knowing the image features that are applied to decide the output of the deep learning algorithms is not possible. The unique features used by the deep learning algorithm to differentiate COVID-19 from CAP cannot be sufficiently visualized by the heatmap, although the heatmap is used to visualize region in images that led to the algorithm output (Li et al., 2020b). Images, especially X-rays and CT scans, are heavily relied on in detecting COVID-19. We suggest that researchers propose explainable deep learning algorithms for the detection of COVID-19 to instill transparency and interpretation in deep learning algorithms.

Misdiagnosis: The application of a deep learning algorithm to detect COVID-19 on a chest CT scan has the possibility of misdiagnosis (Wu et al., 2020) because of the similarity of the COVID-19 symptoms with other types of pneumonia (Belfiore et al., 2020). Incorrect diagnoses can mislead the health professional in deciding and lead to inappropriate medication, further complicating the health condition of the patient with COVID-19. We suggest that researchers combine the CT scan diagnosis using deep learning algorithm with clinical information such as the nucleic acid detection results, clinical symptoms, epidemiology, and laboratory indicators to avoid misdiagnosis (Wu et al., 2020).

Resource allocation: Resource allocation is a challenge as the COVID-19 pandemic keeps spreading because the increase in the number of patients means more resources are required to take care of them. The allocation of limited resources in a rapidly expanding pandemic entails a difficult decision for the distribution of scarce resources (Jiang et al., 2020). The epicenters of the COVID-19 are challenged with resource problems of shortage of beds, gowns, masks, medical staff, and ventilators (Ahuja, Reddy & Marques, 2020). We propose the development of a machine learning decision support system to help in crucial decisions on resource allocation.

Conclusions

In this study, we propose a survey, including a bibliometric analysis of the adoption of machine learning, to fight COVID-19. The concise summary of the projects that adopted machine learning to fight COVID-19, sources of COVID-19 datasets, new comprehensive taxonomy, synthesis and analysis, and bibliometric analysis is presented. The results reveal that COVID-19 diagnostic tools received the most considerable attention from researchers, and energy and resources are more dispensed toward automated COVID-19 diagnostic tools. By contrast, COVID-19 drugs and vaccine development remain grossly underexploited. The algorithm predominantly utilized by the researchers in developing the diagnostic tool is CNN mainly from X-rays and CT scan images. The most suitable CNN architecture for the detection of COVID-19 from the X-ray and CT scan images is ResNet. The challenges hindering practical work on machine learning to fight COVID-19 and a new perspective to solve the identified problems are presented in the study. We believe that our survey with bibliometric analysis could enable researchers to determine areas that need further development and identify potential collaborators at author, country, and institutional levels.

Based on the bibliometric analysis conducted on the global scientific research output on COVID-19 disease spread and preventive measures, the analysis results reveal that most of the research outputs were published in prestigious journals with high influence factors. These journals include The Lancet, Journal of Medical Virology, and Eurosurveillance. The bibliometric analysis also shows the focused subjects in various aspects of COVID-19 infection transmission, diagnosis, treatment, prevention, and its complications. Other prominent features include strong collaboration among research institutions, universities, and co-authorships among researchers across the globe.

Machine learning algorithms have many practical applications in medicine, and novel contributions from different researchers are still evolving and growing exponentially in a bid to satisfy the essential clinical needs of the individual patients, as it is the case with its application to fighting the COVID-19 pandemic. As a way forward, we suggest an in-depth machine learning application review that would focus on the critical analysis of the novel coronavirus disease and other related cases of global pandemics.

Additional Information and Declarations

Competing Interests

Author Contributions

Data Availability

The authors declare that they have no competing interests.

Haruna Chiroma conceived and designed the experiments, performed the experiments, analyzed the data, prepared figures and/or tables, authored or reviewed drafts of the paper, and approved the final draft.

Absalom E. Ezugwu conceived and designed the experiments, performed the experiments, analyzed the data, prepared figures and/or tables, authored or reviewed drafts of the paper, and approved the final draft.

Fatsuma Jauro performed the experiments, prepared figures and/or tables, authored or reviewed drafts of the paper, and approved the final draft.

Mohammed A. Al-Garadi performed the experiments, prepared figures and/or tables, authored or reviewed drafts of the paper, and approved the final draft.

Idris N. Abdullahi performed the experiments, prepared figures and/or tables, authored or reviewed drafts of the paper, and approved the final draft.

Liyana Shuib performed the experiments, prepared figures and/or tables, authored or reviewed drafts of the paper, and approved the final draft.

The following information was supplied regarding data availability:

All raw data are contained in the Methodology and in Table 1.

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
