# Peer review of "Early survey with bibliometric analysis on machine learning approaches in controlling COVID-19 outbreaks"

_PeerJ Computer Science, doi:10.7717/peerj-cs.313_

## Round 0.1 · original submission · Major Revisions

This survey represents a bibliometric analysis for the recent machine learning methods to combat the COVID-19 pandemic which became a great issue, globally. The manuscript is of immense significance to researchers in AI field. However, the manuscript received mixed reviews and I believe it needs a major revision before being recommended for acceptance. I invite you to carefully address the comments of Reviewers 2 and 3 in your rebuttal.

·

Basic reporting

No comment

Experimental design

No comment

Validity of the findings

No comment

Additional comments

Machine learning is one important technique tool to fight the COVID-19 pandemic. As done in this manuscript, to give a early survey with bibliometric analysis is necessary and meaningful. It has not been done previously. The contents in this manuscript are rich. The representations are logical and structured. The listed challenges and future research opportunities are insightful.

There are several comments.

1. The title of Section 6. General Discussions is not appropriate. This part should be the main contents or results of this manuscript. I suggest use the title of "Survey and the bibliometric analysis".

2. Some references are missing.
[1] D. Dong, Z. Tang, S. Wang, et al., The role of imaging in the detection and management of COVID-19: a review. IEEE Reviews in Biomedical Engineering, 2020, DOI: 10.1109/RBME.2020.2990959.
[2] F. Shi, J. Wang, J. Shi, et al., Review of Artificial Intelligence Techniques in Imaging Data Acquisition, Segmentation and Diagnosis for COVID-19, IEEE Reviews in Biomedical Engineering, 2020, DOI: 10.1109/RBME.2020.2987975.
[3] Rahaman, Md Mamunur et al. ‘Identification of COVID-19 Samples from Chest X-Ray Images Using Deep Learning: A Comparison of Transfer Learning Approaches’. 1 Jan. 2020 : 1 – 19.

Reviewer 2 ·

Basic reporting

The paper presents a review for COVID19 detection and diagnosis techniques. The manuscript however lacks readability criterion. There are plenty of information that is not required which makes the reader inconclusive about the study. Therefore, it is suggested for authors to reconsider the approach in order to make it more readable and significant for the research community.
The whole manuscript requires a thorough proof reading. There are plenty of punctuation and grammar mistakes. For example sentence on line 83 to 88 is a one big sentence. It needs to be divided into small and simpler sentences. Moreover the “COVID-19” is repeated 9 times in the same sentence.
The structure of the survey presented in Figure 1 need to be more concise. For example what is the difference in data source in second column and the one mentioned in fifth column.
Figure 2. What is exclusion criteria for exclusion
What is significance of bibliographic coupling in fight against COVID 19.
What is difference between a literature survey and bibliometric analysis. How it gives a better prospective than a literature survey as mentioned in line 103.

Experimental design

In the methodology section, there is a lot of redundant information. The keywords are already mentioned in abstract why need to present as a separate section. Similarly, article selection and bibiliometric protocols are irrelevant as the review is about COVID-19 not on data mining techniques. The authors if they want should present a simple criteria like selecting only from indexed journals such as PubMed, Google Scholar, MEDLINE, Science Direct, Springer, and Web of Science databases and just mention the search terms.
Also the theoretical literature explaining the ANN ,CNN and LSTM is widely discussed so no need to add mathematical theory .One section about theoretical background that discusses the application of there machine learning algorithms will be sufficient.
In section 4.1 line 331 what are the classes 4,3, and 2 present?.

Validity of the findings

In section 6 the subsection 6.2 that discusses the trends of publication, bibliometric analysis, authors prolific analysis etc. seems irrelevant. Instead authors should focus on critical analysis of disease itself.
The COVID-19 detection challenges and how authors have dealt them from the presented literature should be discussed as a summary .From the whole presented literature a more precise summary of analysis should be included.

Additional comments

In summary although the authors have prepared a quiet lengthy review article, but there is a lot of information that is not required. As main objective of any survey is to provide readers an up to date literature and a clear comparative picture in terms of results as well as challenges faced and way forward to solve these problems. It is highly recommended to revisit the approach in this paper and follow a simple criteria to present most relevant and useful analysis.

Annotated reviews are not available for download in order to protect the identity of reviewers who chose to remain anonymous.

·

Basic reporting

In introduction, authors should clearly mention the motivation and address it during the paper. Why is the "bibliometric analysis" important? why authors consider this analysis in this paper.

Experimental design

CNN methods did not describe with details.

Validity of the findings

The conclusion is not satisfactory and cannot convince reader that which methodology is suitable for covid dataset. you should provide that more detailed CNN architecture for each dataset.

Additional comments

1- Authors should provide more details in Table 3 such as: CNN Architecture, number of patients, and database (e.g. type of preprocessing, name of institute).

2- There are several CNN method for analysing CT and X-ray scans. Authors must introduce the best architectures in this database or Covid19 case. They only mentioned that CNN methods have a better performance compare to traditional machine learning methods. It is not clear which type of CNNs are suitable for Covid-19.

3- Quality of some Figures are low, Fig.3 and Fig. 14.

4- It is not clear how many cases are used for training and test in Table 3.

5- In conclusion section, I want to read the author's point of view regarding suitable and successful methods in Covid-19 specifically not generally (CNN methods).

---

## Round 0.2 · Minor Revisions

The authors have addressed the reviewers' comments and improved the whole manuscript. I believe the manuscript is ready for acceptance.

Meanwhile, I do have minor comments which I would like the authors to consider in their final version as follows.
- Reduce the Keywords to 5 only, preferable.
- Update the statistics of COVID-19 cases according to the recent release.
- Double-check the language and grammatical errors. I still can see some places which need improvements.

- If possible, update the reference list to include the very recent works.

---

## Round 0.3 · accepted · Accept

The manuscript has been improved and reviewers' comments have been addressed. The manuscript is ready for Accept.